# Multilingual Diversity Improves Vision-Language Representations

**Thao Nguyen**[1]* **Matthew Wallingford**[1] **Sebastin Santy**[1] **Wei-Chiu Ma**[1,2]

**Sewoong Oh**[1] **Ludwig Schmidt**[1] **Pang Wei Koh**[1,2] **Ranjay Krishna**[1,2]

[1]University of Washington [2]Allen Institute for Artificial Intelligence

## Abstract

Massive web-crawled image-text datasets lay the foundation for recent progress in multimodal learning. These datasets are designed with the goal of training a model to do well on standard computer vision benchmarks, many of which, however, have been shown to be English-centric (e.g., ImageNet). Consequently, existing data curation techniques gravitate towards using predominantly English image-text pairs and discard many potentially useful non-English samples. Our work questions this practice. Multilingual data is inherently enriching not only because it provides a gateway to learn about culturally salient concepts, but also because it depicts common concepts differently from monolingual data. We thus conduct a systematic study to explore the performance benefits of using more samples of non-English origins with respect to English vision tasks. By translating all multilingual image-text pairs from a raw web crawl to English and re-filtering them, we increase the prevalence of (translated) multilingual data in the resulting training set. Pre-training on this dataset outperforms using English-only or English-dominated datasets on ImageNet, ImageNet distribution shifts, image-English-text retrieval and on average across 38 tasks from the DataComp benchmark. On a geographically diverse task like GeoDE, we also observe improvements across all regions, with the biggest gain coming from Africa. In addition, we quantitatively show that English and non-English data are significantly different in both image and (translated) text space. We hope that our findings motivate future work to be more intentional about including multicultural and multilingual data, not just when non-English or geographically diverse tasks are involved, but to enhance model capabilities at large.

## 1 Introduction

Today, the predominant pre-training paradigm for vision-language models relies on large quantities of image-text pairs scraped from the web [35, 24]. As raw web data contains a significant amount of noise, automatic data filtering approaches are designed to curate a high-quality subset and maximize the performance of a model trained on this subset on standard computer vision benchmarks (e.g., ImageNet). However, these benchmarks typically only evaluate in English, and many of them have been shown to be geographically biased: for instance, ImageNet images are mostly sourced from North America and Western Europe [43]. Consequently, it is possible that we are designing data curation algorithms that propagate a monolingual bias, i.e., filtered datasets are increasingly dominated by English image-text pairs. In fact, a lot of highly cited work—including CLIP [35], ALIGN [21] and BASIC [33]—relies exclusively on English data. Using more multilingual data for training is often only a deliberate design decision when non-English tasks are involved [45, 15, 26].

---

*Correspondence to `thaottn@cs.washington.edu`.

38th Conference on Neural Information Processing Systems (NeurIPS 2024).

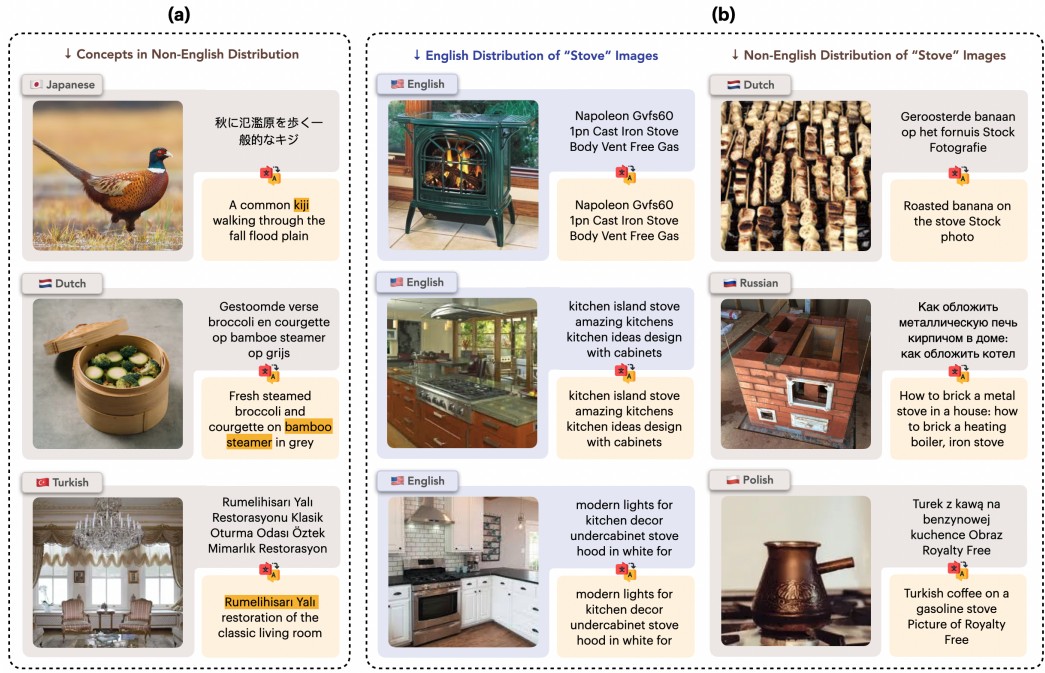

Figure 1: **Multilingual image-text data adds diversity to the English data distribution in various, significant ways** (a) We show some examples of culturally salient concepts that would not exist in "high-quality" English data (as determined by CLIP score), such as "bamboo steamer", "kiji" (the national bird of Japan) and yalı (a traditional architecture style for Turkish waterside houses) (b) Even for a common everyday object ("stove"), non-English and English images portray very different visual representations.

Multilingual data enriches any monolingual data distribution; multilingual data brings attention to culturally salient concepts and introduces new perspectives and annotations for the same visual category [48]. As illustrated in Figure 1, there are certain native concepts, e.g. 'kiji' (the national bird of Japan), that are more likely to be conveyed in Japanese (non-English) captions compared to English ones. Even in the case of a common everyday object ('stove'), the non-English and English images look very different. Despite the diversity present in multilingual data, it is disproportionately excluded from existing large-scale pre-training corpora.

In this paper, we investigate the counterfactual: *can we improve on English vision tasks by diversifying the cultural and linguistic backgrounds of the training data?*

Our investigation is motivated by a dichotomy: English image-text pairs constitute a minority of any random web crawl (in our estimate, one-third); yet, they form a majority in popular pre-training datasets such as LAION-5B [42], DataComp [14], and DFN [13].

It is common for web-scraped corpora to remove "low-quality" data by using a high-performing model (e.g., OpenAI CLIP) to compute image-text alignment and rank the raw data samples. However, this process often disproportionately favors English data if the filtering model also has an English bias [14, 20]. In addition to discarding many potentially useful non-English image-text pairs, this can also negatively impact the geographical and cultural representation of the resulting dataset, and consequently, the model's performance on certain underrepresented populations [38, 36, 10].

Our key observation is that the diversity present in multilingual data can be confounded by the language the data is in, making it difficult to observe the empirical benefits of using such data in model training. To offer a more systematic study of the effectiveness of multilingual data—in contrast to English-only or English-dominated datasets—we fix the language medium, translate all captions from DataComp's 128M-sample web crawl [14] to English with an advanced translation model. We then re-filter the data pool and train a CLIP model on this translated multilingual data. We focus on two types of evaluations: (i) on standard English vision tasks including ImageNet, MSCOCO and Flickr retrieval, and (ii) on geographically diverse images, e.g. from GeoDE [36], which contains images of common objects across different geographical locations. We acknowledge that translation can sometimes be too literal, subject to losing the intent and richness of the original

phrasing. Nevertheless, we hope findings from our work provide a starting point for studying how to leverage the diversity of multilingual data more effectively.

Our contributions are as follows:

- We demonstrate that with translation, non-English data does benefit English vision tasks. In particular, training on more samples of non-English origins leads to better performance on ImageNet, ImageNet distribution shifts and image-English-text retrieval. On the DataComp benchmark with a fixed compute budget, our best-performing approach that leverages translated multilingual captions outperforms training on just filtered raw captions by 2.0% on ImageNet and 1.1 percentage points on average across 38 tasks. When training for longer (which mimics the number of epochs large-scale multimodal models are often trained for), these performance gaps increase to 4.2% and 2.1 percentage points respectively.
- On a geographically diverse task such as GeoDE, training on translated multilingual data leads to 4.2% boost in accuracy on average compared to training on filtered raw data, with performance improvement observed for all regions, especially for Africa where the increase is 5.5%.
- We analyze in detail the differences between English and (translated) non-English image-text pairs. We quantitatively show that they capture distinct distributions, both in text and image space, even after they are converted to the same language medium. Consequently, it is beneficial to combine high-quality data from both sources as much as possible, since they are inherently complementary.

In summary, despite the abundance of "sufficiently useful" English data, existing data curation techniques can always do better in the data diversity axis by being more deliberate about including data from other language and cultural backgrounds. This way of enhancing diversity in turn leads to a better vision-language model *in general*, offering performance benefits beyond non-English vision tasks or tasks involving geographically diverse images. We will release the raw captions and the corresponding English translations for the 128M image-text pairs used in our experiments.

## 2 Related Work

Existing data collection and filtering approaches induce Western bias in downstream datasets and models; benchmarks that seek to capture this bias still receive relatively little attention. Consequently, despite evidence showing cultural and geographical limitations in popular vision datasets, the use of multilingual data is mostly intended for pre-training and fine-tuning multimodal models to do well on non-English tasks. Our work seeks to include more image-text pairs of non-English origins in the pre-training dataset, and shows that this process can improve performance, even on English-centric vision tasks.

**Western bias of existing models and datasets**    Several papers have studied biases in popular datasets, especially biases that correlate with culture and geographic locations. Notably, Shankar et al. [43] find that ImageNet and OpenImages exhibit substantial US-centric and eurocentric representation bias. In NLP, Santy et al. [40] find that existing datasets align predominantly with Western and White populations. It is not only the data collection process that leads to a Western bias, but also the data preprocessing pipeline. For instance, automated data filtering with scores output by a model, e.g., OpenAI CLIP, has been commonly adopted as a way to discard low-quality web-crawled samples. Little is known about the potential biases induced by this approach. Hong et al. [20] recently show that CLIP score filter is more likely to include data related to Western countries compared to that of non-Western countries. In [14] (Figure 24), the authors offer evidence that CLIP filtering implicitly performs some English filtering, as the top CLIP score examples are increasingly dominated by English image-text pairs. Consequently, all these dataset biases translate to performance disparity, as demonstrated by existing work showing that the accuracy of vision systems drops significantly on non-Western inputs [10, 49, 38, 36], or low-resource languages [16].

**Improving the availability of non-English data in multimodal datasets**    Translation has been a popular technique to address the limited availability of large-scale and high-quality non-English data in training and evaluation [45, 47, 6, 32, 11, 2, 16]. In addition to translating English captions into the language of interest, previous work also uses a curated list of common words in the native language to scrape image-text pairs from the web [26, 17]. COCO-CN [25] extends the MSCOCO dataset [7] with manually written Chinese captions.

Most closely related to our setup is the LAION-Translated dataset [29], which translates 3B samples of LAION-5B from many languages into English using the M2M100 model [12]. Compared to this dataset construction, we (i) use a more advanced translation model, NLLB [9], that covers twice as many languages, (ii) work with mostly raw data while LAION was heavily filtered and thus could contain a biased representation of multilingual data. To the best of our knowledge, no existing work has experimented with the LAION-Translated dataset.

**Adapting CLIP post-training for multilingual tasks**  Geigle et al. [15] translate high-quality English data into 95 languages, and use these translated samples to re-align an image encoder previously trained on English data to a multilingual language model. Similarly, Chen et al. [8] propose re-training OpenAI CLIP on a mix of Chinese and English data to enhance its multilingual representation. In [4], the authors explore adaptation without any image data, solely fine-tuning the text encoder with English captions from MSCOCO, Google Conceptual Captions, and VizWiz translated to other languages. Visheratin [45] replace the text encoder of OpenAI CLIP with the text encoder from the NLLB model, and fine-tune the new model on multilingual image-text pairs obtained from translating LAION-COCO's English captions [41] into 200 languages. In contrast to these papers that employ multilingual data for the purpose of adapting to non-English tasks, we focus on using multilingual data to do better on common vision tasks that are in English.

**Using multilingual data significantly enhances data diversity**  Our study is partly inspired by findings from Ye et al. [48], who show that multilingual synthetic captions obtained from existing image captioning systems provide higher semantic coverage than monolingual ones, over 3.6K images. Our experiments instead use raw web-crawled data and explore the performance benefits of embracing cultural and linguistic diversity in (mostly) human-generated captions *at scale*.

## 3  Experimental Setup

Given a starting pool of raw image-text pairs scraped from the web, many of which contain non-English captions, we experiment with ways to preprocess and filter this pool into a high-quality dataset. The quality of the dataset is measured by the zero-shot performance of a CLIP model trained on it from scratch.

**Data**  We experiment with the medium pool of the DataComp benchmark [14], which consists of 128M image-text pairs randomly sampled from Common Crawl dumps between 2014 and 2022, and deduplicated. Unlike other heavily filtered corpora such as LAION [42], DataComp applies minimal data preprocessing, involving only NSFW filtering, deduplication of evaluation sets, and face blurring. This allows the candidate pool to stay close to the natural distribution of the raw web data as much as possible, in addition to enabling maximum flexibility in dataset design.

**Translation model**  To detect language and translate the raw captions from DataComp into English, we use the No Language Left Behind (NLLB) translation model [9], which is considered state-of-the-art. NLLB is the first to translate across 200 languages, including low-resource ones that are not currently supported by common translation tools. We use the 600M-parameter model publicly available on HuggingFace to allow for fast inference on our large data corpus. All 128M captions from DataComp are translated to English; examples could be found in Appendix A. We provide some quantitative analysis of the translation quality in Appendix C.

**Training**  After translating the captions of all samples in the raw data pool, we filter them based on cosine similarity between image and text embeddings. We experiment with using OpenAI CLIP-ViT-L/14 [35] and the *public* Data Filtering Network (DFN) from [13] to obtain the embeddings, and subsequently, the cosine similarities. The DFN, specifically designed to filter data for subsequent model training, was trained on three public datasets deemed as high-quality—Conceptual Caption 12M [5], Conceptual Captions 3M [44], and Shutterstock 15M [30]. We find that indeed the public DFN is better at data filtering compared to OpenAI CLIP, as measured by the performance of CLIP trained on the corresponding filtered datasets (see Appendices E and G).

We pretrain a CLIP model [35] on each filtered subset with ViT-B/32 as the image encoder, and follow DataComp's hyperparameters; details can be found in Appendix B. Unless specified otherwise, all models are trained for the same compute budget (128M steps) as determined by DataComp. For some

Table 1: **On the DataComp benchmark, training on translated captions outperforms training on raw captions across a range of metrics; using both types of captions yields even more performance gains.** We report the performance of select baselines on the DataComp benchmark [14]; all baselines are trained for the same number of steps as specified. Here the filtering threshold (and thus the resulting dataset size) has been tuned for each baseline and we only show the filtered subset that yields the highest average accuracy. We find that with the same filtering method (i.e., using DFN score), training on translated captions ("Filtered translated captions") is more effective than training on raw captions ("Filtered raw captions") as seen from higher performance on ImageNet, ImageNet distribution shifts, retrieval, GeoDE (worst-region accuracy) and on average across 38 tasks. Combining both sources of captions leads to the best performance. Appendix G contains the full results.

| Baseline name | Dataset size | ImageNet | ImageNet shifts | Retrieval | GeoDE | Average over 38 tasks |
|---|---|---|---|---|---|---|
| *Training with DataComp setup (128M steps)* | | | | | | |
| Filtered raw captions | 25.6M | 0.316 | 0.260 | 0.282 | 0.688 | 0.350 |
| Filtered raw captions, replaced with translated captions | 25.6M | 0.304 | 0.252 | 0.268 | 0.668 | 0.331 |
| Filtered translated captions | 25.6M | 0.329 | 0.275 | 0.296 | 0.709 | 0.359 |
| Filtered English-only captions | 16.0M | 0.283 | 0.236 | 0.278 | 0.666 | 0.327 |
| Filtered raw captions ∪ Filtered translated captions | 34.2M | 0.329 | 0.271 | 0.298 | 0.720 | **0.364** |
| Filtered raw captions & Filtered translated captions | 51.2M | **0.336** | **0.280** | **0.301** | **0.725** | 0.361 |
| *Training for 10× longer (1.28B steps)* | | | | | | |
| Filtered raw captions | 38.4M | 0.414 | 0.340 | 0.344 | 0.742 | 0.414 |
| Filtered translated captions | 38.4M | 0.427 | 0.347 | 0.352 | 0.771 | 0.414 |
| Filtered raw captions ∪ Filtered translated captions | 34.2M | 0.441 | 0.359 | 0.353 | 0.775 | 0.427 |
| Filtered raw captions & Filtered translated captions | 51.2M | **0.456** | **0.369** | **0.371** | **0.776** | **0.435** |

select baselines, we also experiment with training for 10× longer. The fixed architecture, compute and hyperparameter setup allow us to isolate data quality as the main factor influencing performance.

**Evaluation** We perform zero-shot evaluation of trained CLIP models using the 38 tasks from DataComp. These tasks involve recognition and classification of a wide range of domains (e.g., texture, scene, metastatic tissue, etc.) in addition to image-text retrieval and commonsense association. Among them, we pay particular attention to commonly cited metrics such as ImageNet accuracy, ImageNet distribution shift accuracy - a proxy for natural robustness, and retrieval performance. ImageNet shifts include ImageNet-V2 [37], ImageNet Sketch [46], ImageNet-A [19], ImageNet-O [19], ImageNet-R [18] and ObjectNet [1]. Retrieval score is the average of the performance on Flickr30K [50], MSCOCO [7] and WinoGAViL [3]. Throughout the paper we also highlight GeoDE worst-region performance [36]—a task that involves geographically diverse images—to demonstrate the added benefits of geographical inclusivity that training on more (translated) multilingual captions offers.

## 4 Impacts of using (translated) multilingual captions on standard vision tasks

We explore training on each caption distribution separately, as well as combining them. Below we describe the baselines from Table 1 in more detail:

- *Filtered raw captions:* As mentioned in Section 3, we use the *public* DFN from [13] by default to filter the starting pool (128M samples). Given the images and the corresponding web-crawled captions, we experiment with varying the filtering threshold to keep top x% of the pool based on DFN score. In Table 1, we only report the best average performance obtainable after the filtering threshold has been tuned, and the resulting dataset size. Refer to Appendix G for the full results.
- *Filtered translated captions:* Similar to the approach above, we tune the filtering threshold, but using DFN score between an image and the English translation of the original web-crawled caption.
- *Filtered English-only captions:* Similar to "Filtered raw captions" baseline, here we also tune the filtering threshold to keep only a subset of the pool with the highest DFN scores, but with an additional constraint of only filtering from samples with web-crawled captions already in English.

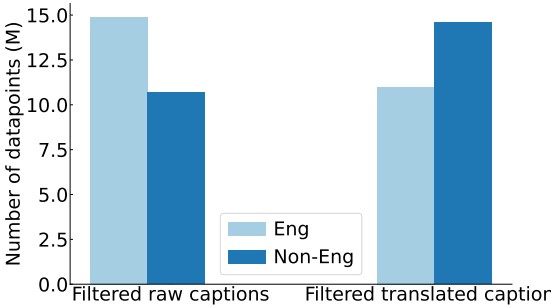

Figure 2: **Filtering with translated captions allows substantially more (translated) non-English samples to be included in the final training set.** While English data only makes up about one-third of the raw web crawl, it dominates the top 20% of the pool, selected based on DFN score between image and *raw* caption. With translation, English-translated non-English captions now make up the majority of the "high-quality" data and thus are more likely to be selected for training.

- *Filtered raw captions, replaced with translated captions:* Given samples from "Filtered raw captions" (i.e. again, based on the cosine similarity score between image and original web text), we keep the images selected and replace the raw captions with the corresponding English translations.
- *Filtered raw captions ∪ Filtered translated captions:* We combine image-text pairs from "Filtered raw captions" and "Filtered translated captions" subsets uncovered above. However, these subsets have about two-thirds of their images in common (see Appendix D). For such images, we only include one copy in the final training set and use English-translated caption by default. For the rest of the images in "Filtered raw captions" that do not appear in "Filtered translated captions", we include them in the training set with the corresponding original captions (which could be non-English).
- *Filtered raw captions & Filtered translated captions:* We combine "Filtered raw captions" and "Filtered translated captions"; the overlapping images between these two subsets would appear twice in the final training set - one copy with the original web caption and one copy with the English-translated caption.

## 4.1 Overall performance trends

**Combining high-quality raw and translated captions offers the best performance** We find that using *both* sources of captions, and image data—since top (image, raw text) pairs and top (image, translated text) pairs only have two-thirds of the images in common—leads to the best performance (bolded entries of Table 1). This approach surpasses training on only high-quality raw data by 2% on ImageNet, ImageNet shifts and retrieval, and improves GeoDE worst-region performance by 3.7%. We note that this is not simply due to having more unique image-text samples, as filtered subsets of similar sizes but using a single source of captions (e.g., top 40% raw captions totalling 51.2M samples) yields significantly lower performance (Appendix G).

**Using only translated multilingual captions is still better than using only raw captions** Zooming in on "Filtered translated captions" and "Filtered raw captions" baselines, we find that the former outperforms the latter on many standard metrics (ImageNet, ImageNet distribution shifts, image-text retrieval). This is unexpected in light of prior work showing that ImageNet exhibits strong amerocentric and eurocentric bias [43], with images from America and Great Britain taking up 53% of the dataset.

## 4.2 Ablations

We perform more ablation studies to disentangle the reasons for the performance gains from using high-quality translated captions, as well as to verify that the gains are robust.

**The performance gain from using translated captions is not simply due to converting all text data to a common language medium** Given image-text pairs from the "Filtered raw captions" subset, we replace the web-crawled captions with the corresponding English translations ("Filtered raw captions, replaced with translated captions"). This intervention on only the captions leads to performance drop across the board. We hypothesize that this is due to noise in the translation process, as (i) many web captions are formed by stringing together short, ungrammatical phrases and thus are "out-of-distribution" for the NLLB translation model, (ii) web captions may contain multiple languages in the same sentence, thereby leading to noisy language detection and translation.

**Re-filtering data after translation is also necessary due to significant changes in the data ranking** Besides noisy artifacts introduced by translation, the process also changes the image-text cosine

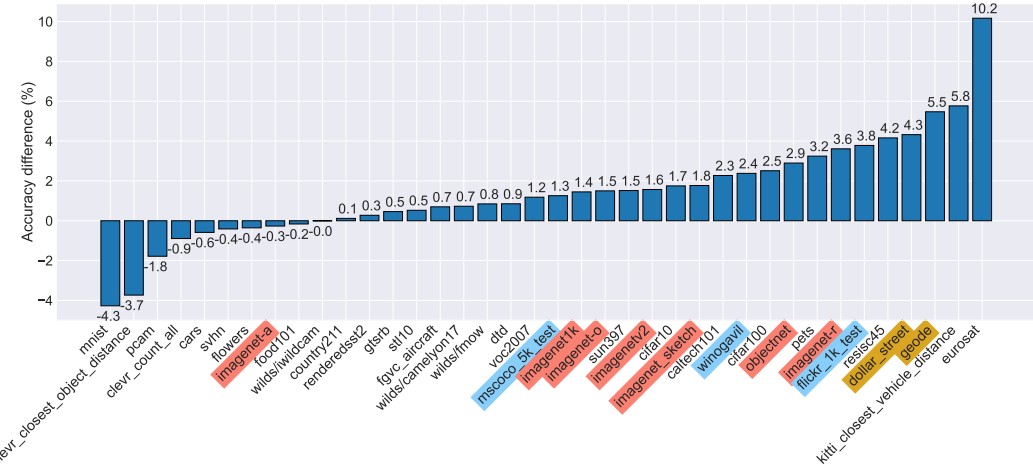

Figure 3: **With the same degree of filtering, training with (image, translated caption) pairs improves performance on 28 out of 38 tasks compared to training with (image, raw caption) pairs, including on ImageNet distribution shifts, retrieval, and tasks with geographically diverse inputs.** We compare performance on each task of the DataComp benchmark between training with raw captions and training with translated captions. Both datasets have been filtered with image-text cosine similarities output by the public DFN [13] to select the top 30% examples. We find that using translated captions leads to 1.5 percentage points improvement on average across 38 tasks. We highlight the performance changes on ImageNet distribution shifts (**red**), retrieval (**blue**) and fairness-related tasks (**dark yellow**).

similarity score and thus the quality ranking of the data samples in the pool. More specifically, we find that while "Filtered raw captions" is dominated by English samples, (translated) non-English samples make up the majority of "Filtered translated captions" (Figure 2). These two filtered subsets only share about two-thirds of the images in common, see Appendix D for more details. Therefore, by changing the caption distribution, we are also inducing changes to the image distribution that the best-performing model would see.

**The benefits of training with translated multilingual captions are consistent across data filtering networks** As alluded to in Section 3, we also explore using cosine similarities output by OpenAI CLIP-ViT-L/14 [35] for data filtering. The full results for this ablation can be found in Appendix E. Similar to the previous observations, we find that using filtered translated captions yields better performance than using filtered raw captions.

**The performance benefits of using translated data persists with much longer training duration** We also experiment with training for $10\times$ more steps (i.e., 1.28B samples seen) as this is more in line with the number of epochs typical vision-language models are often trained on (e.g., OpenAI CLIP models were trained on 400M datapoints for 32 epochs [35]). When using either the raw caption or the translated caption distribution, setting the filtering threshold to top 30% of the pool works best. Even though the two filtered datasets now yield the same average accuracy, training on high-quality translated captions still offers significant advantages when it comes to ImageNet, ImageNet shifts, retrieval and GeoDE. Combining high-quality data from both sources of captions continues to be the best performing approach, giving 4.2% improvement on ImageNet and 2.1 percentage points improvement on average, compared to just training on filtered raw captions. Results for more baselines can be found in Appendix H.

### 4.3 Individual task analysis

After observing improvement across different metrics from using more (translated) multilingual captions, in this section we break down the performance changes for each of the 38 tasks in the DataComp benchmark. The base model for comparison is CLIP trained on top 30% image-text pairs filtered from the raw data pool, and the new improved model is the one trained on top 30% image-text pairs after the same pool has been all translated to English. Both models are trained for 128M steps. Averaged across 38 evaluation tasks, the latter yields a 1.5 percentage points improvement. The biggest gains come from Flickr retrieval, fairness (GeoDE, Dollar Street) and remote sensing (EuroSAT, RESISC45) tasks (Figure 3).

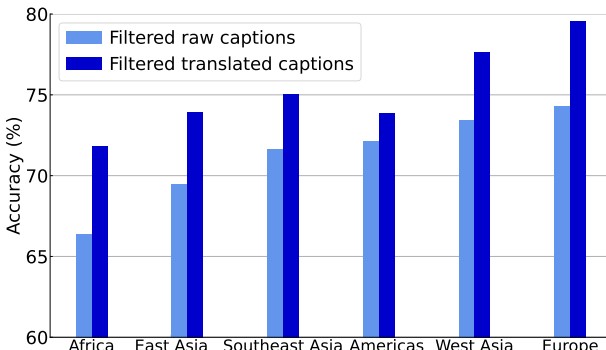

Figure 4: **On GeoDE, using filtered translated captions leads to improvements across *all* regions compared to using filtered raw captions, with Africa observing the biggest gain.** We break down the GeoDE performance by region and compare training on top 30% translated captions to training on top 30% raw captions. On average, classification accuracy improves by 4.2%, and the improvement applies to all regions in the dataset, especially Africa where the accuracy gain is the biggest at 5.5%.

English data                Non-English data

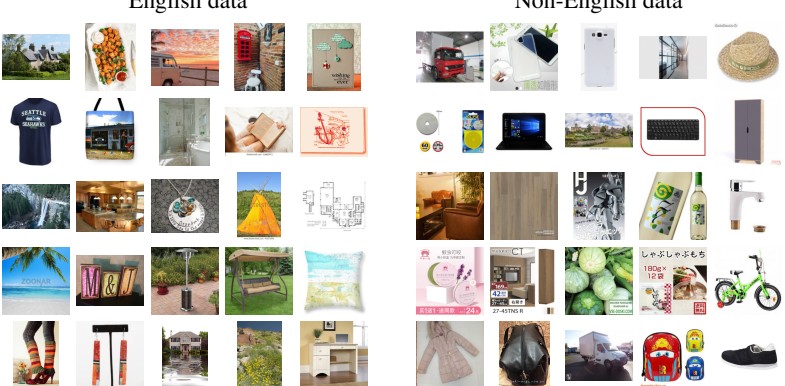

Figure 5: **Visualizations of what an SVM deems typical of images with English captions and those with non-English captions.** We show examples of easy-to-classify images in our English versus non-English data classification task. Besides the product logo and text in some images that are suggestive of the language distribution, the image content mostly depicts common scenes and objects.

In particular, on GeoDE [36], which consists of images of common objects crowd-sourced from six different regions across the world, we find that using translated multilingual captions makes CLIP perform better on *all* regions, with the biggest gain coming from Africa images (5.5%) and the second biggest gain coming from the Europe region. This is unexpected given that African languages only make up a small fraction of the training set, and after translation, more European (compared to African) language samples make it to the resulting filtered subset (Appendix D). Besides, it is worth noting that on GeoDE, our best baseline from Table 1 ("Filtered raw captions & Filtered translated captions") outperforms the current best baseline on DataComp's medium scale ("HypeSampler" [22]) by 1.5%, measured in terms of worst-region accuracy (Africa).

## 5 Understanding the differences between English and (translated) non-English data

Given that using more image-text pairs of non-English origins in the training set offers significant benefits on most vision tasks, including those that are English-centric, we seek to further understand the various ways that non-English data complements and improves the diversity of English data, in both image and text space.

### 5.1 Image distribution

As a proxy for capturing image distribution differences, we train simple classifiers—a Support Vector Machine (SVM) on CLIP embeddings and a ResNet-50—to distinguish images with English captions from those with non-English captions. We randomly sample 100K images from each distribution for training and 10K for testing. We only use images from the top 20% of the candidate pool (based on DFN cosine similarity score), to ensure that (i) these are the samples that the best-performing CLIP models are eventually trained on, (ii) images are of sufficient quality, to the extent that they have fitting captions accompanying them.

We note that this classification task is non-trivial for a number of reasons:

- Many images are duplicated across the web, i.e., after DataComp performs image deduplication it is possible for these images to appear with either English captions or non-English captions in our data pool.
- The language detection model is not perfect and web-crawled captions may contain more than one language in the same sentence.
- Images with non-English captions contain many sub-distributions of images, some of which may overlap with the distribution of images with English captions (e.g., eurocentric data).

Despite these challenges, our simple classifiers achieve 67% accuracy on the binary classification task, significantly better than random chance performance. In Figure 5, we show some examples of what the SVM deems easy to classify. Overall this experiment suggests that the distribution of images with non-English captions is sufficiently distinct from that of images with English captions. Therefore, not training on more of the former means we are missing out on a considerable amount of visual information that can only be found in a separate part of the web.

| Text distributions | MAUVE score |
|---|---|
| Translated non-English vs. Translated non-English | $0.964 \pm 0.005$ |
| Translated English vs. Translated English | $0.957 \pm 0.005$ |
| English vs. Translated English | $0.890 \pm 0.004$ |
| Translated English vs. Translated non-English | $0.616 \pm 0.010$ |
| English vs. Translated non-English | $0.449 \pm 0.008$ |

Table 2: **There exists a substantial gap between the distribution of English captions and that of non-English captions, even when we apply translation to both, suggesting that they capture different contents.** We use MAUVE score [34] to measure the difference between English captions and (translated) non-English captions in the training set. We find that (i) translation indeed introduces some artifacts and changes what "English" texts may look like, (ii) the English text distribution is remarkably different from the non-English one, even after they are converted to the same medium with translation. All scores are averaged over 3 randomly sampled sets of 10K captions.

## 5.2 Text distribution

In the text space, we leverage MAUVE score [34] to quantify the differences between English captions and non-English captions that have been translated to English. MAUVE was originally designed to measure the gap between machine- and human-generated texts. The metric computes KL divergences in a quantized, low-dimensional space after embedding text samples from each distribution with a language model (by default, GPT-2). The output score ranges between 0 and 1 and the higher it is, the more similar the text distributions are. Similar to our analysis in the image space, we only use caption samples from the top 20% of the candidate pool (based on DFN score).

In Table 2, as a sanity check, we randomly sample two disjoint sets of 10K captions from the same text distribution (e.g., non-English captions having been translated to English with the NLLB translation model, *or* English captions having been passed through the same model). We find that the two sets indeed exhibit high MAUVE scores (above 0.95). When comparing raw English captions to English captions that have been passed through the NLLB model (i.e., "translated English"), we find that the MAUVE score decreases slightly (0.890), indicating that the translation process introduces some artifacts making English-translated English text look somewhat different from raw English text.

When comparing raw English texts and non-English texts, both having been passed through the translation model and thus undergone the same "preprocessing" (i.e., English translation), the resulting MAUVE score is relatively low (0.616). This signals that independent of differences in language, what is discussed in English captions and non-English captions differs in many ways. We should therefore leverage both sources of text information as much as possible for training.

## 6 Discussion

**Limitations** We fix the data filtering method to be based on image-text cosine similarity output by a trained model, and study the impact of selecting training data based on different caption distributions. We show that the advantages of using translated multilingual data are robust to the choice of the filtering network. However, our best-performing baseline is currently not state-of-the-art on the DataComp benchmark [14]. It remains an open question whether the performance benefits of our method persist with other score metrics, e.g. hyperbolic entailment [22] - currently the best method for DataComp's medium scale, or other filtering methods that are used jointly with CLIP score, e.g. T-MARS [27] which also removes text-spotting images with limited visual information.

Besides, we acknowledge that translation can introduce artifacts and reduce the richness of expressions in the original languages. Prior work has shown that translated sentences are less effective compared to manually-written sentences as sources of training data for a vision task, e.g. image captioning [28, 23]. Our work mainly leverages translation as a way to convert all image-text pairs to the same medium, and remove confounding impacts of language in data selection and model training.

**Conclusion**   In this work, we bring all web-crawled image-text pairs into a common language medium via English translation, and systematically study the empirical benefits of using non-English data with respect to standard computer vision tasks (that are in English). By including significantly more (translated) multilingual data in the filtered training set, the improved cultural and linguistic diversity in turn leads to substantial gains across all major metrics—ImageNet, distribution shift robustness, retrieval capabilities and average performance across 38 tasks—even if some of these metrics have been shown to overfit to English. We also find that despite being translated to the same language, English and non-English data distributions are still distinct from each other.

**Future work**   This work motivates future studies into data curation techniques that directly improves the diversity of data origins. Another interesting direction of exploration is adapting trained CLIP models from this paper for multilingual benchmarks, such as by re-training the text encoder (that has only been trained on English and English-translated captions) with the technique proposed in [4]. We hypothesize that text adaptation alone is sufficient for our models to perform competitively on non-English tasks, owing to the presence of significantly more multilingual and multicultural images in our pre-training dataset.

**Broader impact**   While most studies have looked into non-English data with the goal of increasing societal representation and subsequently improving performance on under-served populations or tasks, we observe that non-English data can actually help enhance model capabilities *as a whole* (including on standard English benchmarks). This suggests that diverse representation in training data, e.g. as measured by cultural and linguistic backgrounds, should be a deliberate design decision in the data curation process, instead of existing only as a byproduct of the preprocessing pipeline or out of societal considerations.

## Acknowledgments and Disclosure of Funding

We thank Stability AI for the assistance with compute resources. We are grateful to Rachel Hong, Jonathan Hayase, Cheng-Yu Hsieh, Anshul Nasery and Amita Kamath for offering feedback on the manuscript and other helpful discussions. TN is supported by the UW-Meta AI Mentorship Program. This work is supported in part by the Sony Faculty Innovation grant, NSF awards 2019844, 2134012, and 2112471, NSF AI Institute for Foundations of Machine Learning (IFML), Microsoft Grant for Customer Experience Innovation, Open Philanthropy, and the Allen Institute for AI. PWK is supported by the Singapore National Research Foundation and the National AI Group in the Singapore Ministry of Digital Development and Innovation under the AI Visiting Professorship Programme (award number AIVP-2024-001).

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

# A    Examples of translated data (no cherry picking)

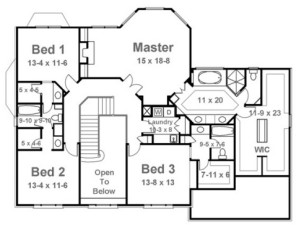

*Raw caption:* Lovely 2nd Floor Plans Part - 4: 2nd Floor Plan
*Language detected:* eng_Latn
*Translation:* Lovely 2nd Floor Plans Part - 4: 2nd Floor Plan

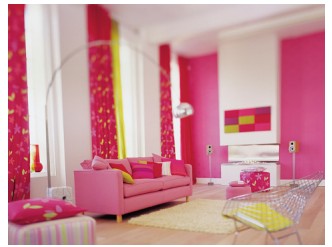

*Raw caption:* ピンク色「インテリアの鮮やかなピンクのカ ラフルなラウンジ」:スマホ壁紙(19)
*Language detected:* jpn_Jpan
*Translation:* Pink: The bright pink lounge of the interior: cell phone wallpaper.

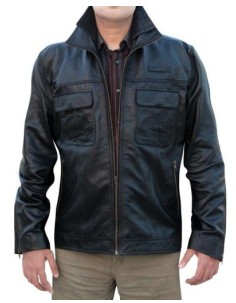

*Raw caption:* CW’s The Originals Joseph Morgan Jacket
*Language detected:* eng_Latn
*Translation:* CW The Originals Joseph Morgan Jacket

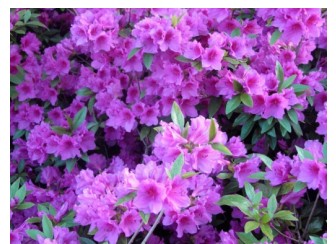

*Raw caption:* PURPLE AZALEAS UP CLOSE
*Language detected:* yue_Hant
*Translation:* Purple Azleas up close.

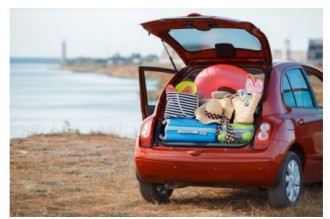

*Raw caption:* Paso a paso: Cómo sacar el permiso para circular con el auto en vacaciones de verano | Garantia Plus
*Language detected:* spa_Latn
*Translation:* Step by step: How to get a driving permit on summer vacation

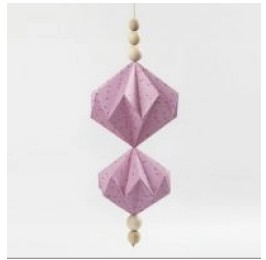

*Raw caption:* Een hangende decoratie van Vivi Gade papieren diamantvormen
*Language detected:* nld_Latn
*Translation:* A pending decoration of Vivi Gade paper diamond shapes

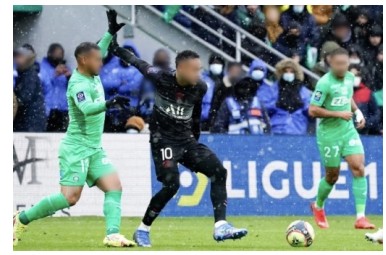

*Raw caption:* Neymar n'a plus joué en compétition depuis le 28 novembre 2021, à Saint-Etienne. Icon Sport
*Language detected:* fra_Latn
*Translation:* Neymar has not played in a competitive match since 28 November 2021, at Saint-Etienne.

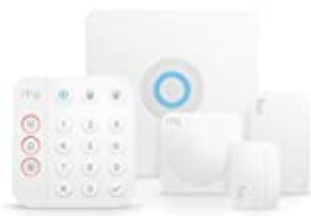

*Raw caption:* Ring Alarm 5-piece kit (2nd Gen) – home security system with optional 24/7 professional monitoring – Works with Alexa
*Language detected:* eng_Latn
*Translation:* Ring Alarm 5-piece kit (2nd Gen) home security system with optional 24/7 professional monitoring Works with Alexa

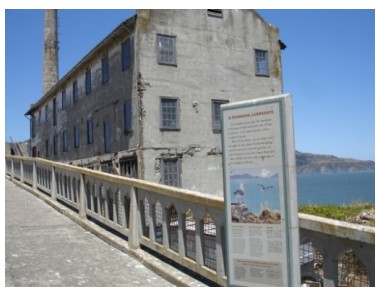

*Raw caption:* Alcatras Hapisanesi
*Language detected:* tur_Latn
*Translation:* The Alcatras Prison

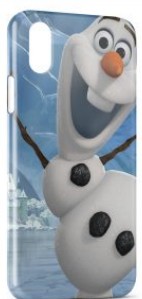

*Raw caption:* Coque iPhone XS Max Olaf Reine des neiges bonhomme de neige
*Language detected:* fra_Latn
*Translation:* Iphone XS Max Olaf Snow Queen Snowman

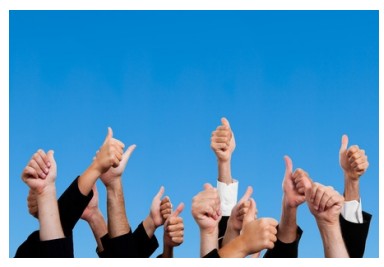

*Raw caption:* Multiracial Thumbs Up Against Blue Sky
*Language detected:* eng_Latn
*Translation:* Multiracial Thumbs Up Against Blue Sky

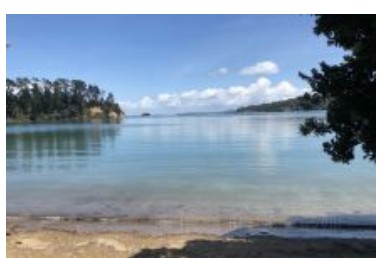

*Raw caption:* 去年学校野去了waiheke island 景色超美 海水十分清澈
*Language detected:* deu_Latn
*Translation:* Last year school camp went to Waiheke island

## B   More training details

We follow the training and evaluation protocols of the DataComp benchmark [14]; refer to Appendices M and N of this previous work for more details. To summarize, we use ViT-B/32 as the image encoder for CLIP, and fix the hyperparameters used for training: learning rate 5e-4, 500 warmup steps, batch size 4096, AdamW optimizer $\beta_2 = 0.98$.

Since the compute budget is fixed, for the DataComp setting (128M training steps), each of our baseline takes about 8 hours with 8 A40 GPUs and 40 CPUs. With the same amount of resources, for experiments involving training for longer (1.28B steps), each baseline takes about 80 hours. We report all baselines that we ran in Appendices E, G and H.

## C   Translation quality

Here we provide a quantitative assessment of the quality of caption translation offered by the NLLB model [9]. We sample 100K captions from the raw data pool and backtranslate the English-translated caption into the original (detected) language (e.g., Chinese text → English translation → Chinese translation of the English-translated text). To evaluate the translation quality, we compute the cosine similarity between the initial web-scraped text and the backtranslated text using embeddings from the multilingual Sentence-BERT model [**?** ]. We find that on average the cosine similarity (and thus, translation quality) remains relatively high (0.63). In the table below, we report the top 5 and bottom 5 languages that observe the highest and lowest translation quality as captured by our metric, computed over at least 30 text samples per language.

Table 3: Top 5 and bottom 5 languages where web-scraped captions observe the highest and lowest translation quality by the No Language Left Behind model [9], out of all the languages detected in our raw data pool. Translation quality is measured by how much the semantic meaning is preserved after the caption is translated into English and subsequently backtranslated into the original language.

| Language | Text cosine similarity after backtranslation (↑) |
|---|---|
| English | 0.886 |
| Norwegian Nynorsk | 0.883 |
| Bengali | 0.883 |
| Russian | 0.860 |
| Norwegian Bokmål | 0.839 |
| Marathi | 0.271 |
| Irish | 0.240 |
| Standard Latvian | 0.233 |
| Chechen | 0.0595 |
| Karachay-Balkar | 0.00280 |

## D   Changes in data composition due to translation

### D.1   Differences in image uids between "Filtered raw captions" and "Filtered translated captions"

In this section, we provide some statistics of the differences in image-text pairs selected for "Filtered raw captions" and "Filtered translated captions", when both caption distributions are filtered to a similar extent with the public DFN from [13]. We find that at either 20% or 30% selectivity threshold, both filtered subsets have about two-thirds of their images in common. In addition, filtering the initial pool using (image, translated caption) cosine similarity score always leads to translated multilingual captions taking up the majority of the resulting training set. This is not the case when filtering with (image, raw caption) cosine similarity score.

Table 4: Analysis of the number of samples of English and non-English origins in "Filtered raw captions", "Filtered translated captions" and their intersection.

| Data subset | Total size (M) | Number of English captions (M) | Number of non-English captions (M) |
|---|---|---|---|
| Top 20% Raw captions | 25.6 | 14.9 | 10.7 |
| Top 20% Translated captions | 25.6 | 11.0 | 14.6 |
| Top 20% Raw captions ∩ Top 20% Translated captions | 17.1 | 10.7 | 6.4 |
| Top 30% Raw captions | 38.4 | 20.4 | 18.0 |
| Top 30% Translated captions | 38.4 | 15.4 | 23.0 |
| Top 30% Raw captions ∩ Top 30% Translated captions | 25.7 | 15.0 | 10.7 |

## D.2   Language composition of the filtered subsets

Below we show the most common languages in top 20% raw captions and top 20% translated captions.

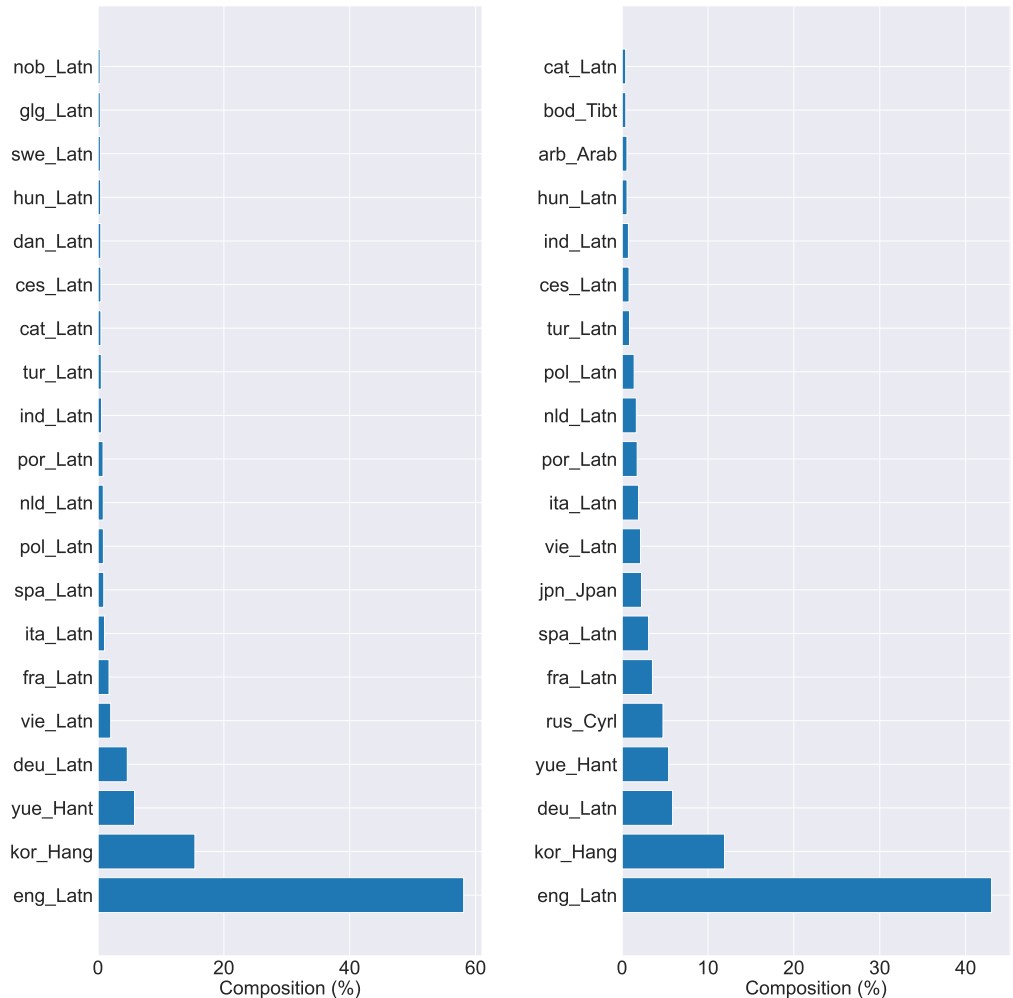

Figure 6: Top 20 languages that are most common in top 20% raw captions (left) and top 20% translated multilingual captions (right) (both are filtered with the public DFN model).

### D.3 Changes in language composition

Comparing image-text pairs selected in top 20% translated captions to those selected in top 20% raw captions, we show below the languages that observe the biggest change in their representation in the resulting training set:

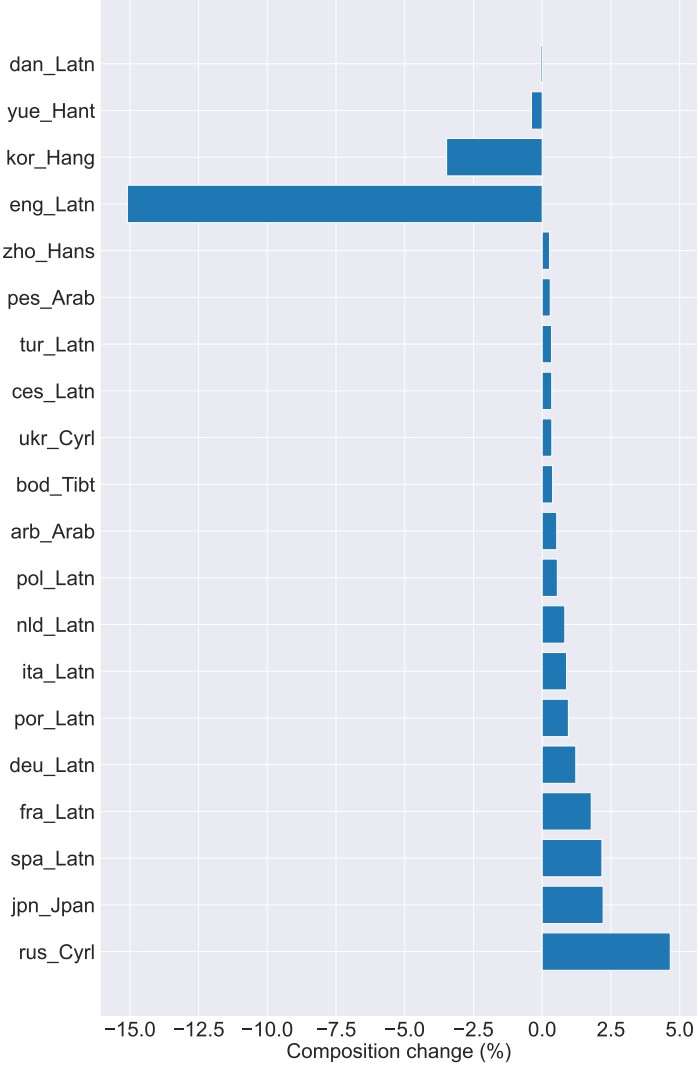

Figure 7: Top languages that see the biggest change (in absolute percentage) in their representation in the final training set when we filter with translated multilingual captions versus with raw web-scraped captions.

# E Experiments with OpenAI CLIP score filtering

Table 5 shows results of our experiments with data filtering using OpenAI CLIP-ViT-L/14 [35]. Besides the baselines described in Section 4, the table also contains "Top 30% raw captions ∪ top 30% translated captions, using translated caption for all", where we take all the images uncovered from "Top 30% raw captions" and "Top 30% translated captions", deduplicate them, and use the corresponding translated captions for all these images.

We find that with OpenAI CLIP as the filtering network, some of our observations from Section 4 continue to hold true: (i) using translated multilingual captions is better than using raw captions, and (ii) the performance gain from training with translated captions requires re-filtering the entire data pool after translation (as seen from comparing the first two baselines of the table).

Table 5: **The benefits of using translated multilingual captions still hold when using cosine similarity score from OpenAI CLIP for filtering.** This table shows performance of all baselines we experiment with for filtering with OpenAI CLIP-ViT-L/14. Again, the compute budget is fixed and all baselines are trained for 128M steps. We find that training on filtered translated captions also outperforms training on filtered raw captions in this case.

| Baseline name | Dataset size | ImageNet | ImageNet shifts | Retrieval | GeoDE | Average over 38 tasks |
|---|---|---|---|---|---|---|
| Top 30% raw captions | 38.4M | 0.273 | 0.230 | 0.251 | 0.683 | 0.328 |
| Top 30% raw captions, replaced with translated captions | 38.4M | 0.260 | 0.224 | 0.248 | 0.660 | 0.322 |
| Top 30% translated captions | 38.4M | **0.292** | **0.250** | 0.267 | 0.695 | **0.342** |
| Top 50% raw captions | 64.1M | 0.254 | 0.218 | 0.262 | 0.670 | 0.315 |
| Top 50% translated captions | 64.1M | 0.265 | 0.230 | **0.276** | **0.704** | 0.320 |
| Top 30% raw captions ∪ top 30% translated captions, using translated caption for all | 47.7M | 0.275 | 0.234 | 0.261 | 0.683 | 0.326 |
| Top 30% raw captions ∪ top 30% translated captions | 47.7M | 0.284 | 0.247 | 0.260 | 0.696 | 0.340 |
| Top 30% raw captions & top 30% translated captions | 76.8M | 0.289 | **0.250** | 0.262 | 0.696 | 0.335 |

# F    Comparison to training with synthetic captions

Table 6: **When fixing the training images and replacing translated English captions with synthetic captions generated by BLIP2, we find that performance decreases in general.** Since filtering from translated captions exposes CLIP to both new images and new text distributions, we seek to disentangle the impact of these two factors on model performance. Our results suggest that having access to more diverse images alone (without the corresponding translated multilingual captions) may be insufficient for achieving performance gains.

| Baseline name | Dataset size | ImageNet | ImageNet shifts | Retrieval | GeoDE | Average over 38 tasks |
|---|---|---|---|---|---|---|
| Top 20% translated captions | 25.6M | 0.329 | 0.275 | 0.296 | 0.709 | 0.359 |
| Top 20% translated captions, replaced with synthetic captions | 25.6M | 0.283 | 0.255 | 0.350 | 0.696 | 0.336 |
| Top 30% translated captions | 38.4M | 0.311 | 0.265 | 0.305 | 0.718 | 0.351 |
| Top 30% translated captions, replaced with synthetic captions | 38.4M | 0.282 | 0.253 | 0.371 | 0.703 | 0.341 |

As observed in Section 4.1, training on filtered translated captions outperforms training on filtered raw captions across all major metrics. This could be attributed to both changes in captions (from original web-crawled texts to English-translated texts) as well as changes in images (since "Filtered raw captions" and "Filtered translated captions" only share some of the images in common, see Appendix D). Here we attempt to disentangle the contribution to performance gain from these two changes.

Given the images selected by filtering based on (image, translation caption) cosine similarity (i.e., "Top 20% translated captions", "Top 30% translated captions"), we generate synthetic captions for each image using BLIP2 model [24] and the generation hyperparameters from [31]. Training on the new (image, synthetic caption) pairs leads to lower performance overall compared to training on the original (image, translated caption) pairs (Table 6). This suggests that having access to more diverse (non-English) images in the training set is not sufficient to boost performance; the diversity coming from translated multilingual captions is also necessary for obtaining the accuracy gains.

We acknowledge that since BLIP2 was pre-trained on relatively few multilingual samples, it is possible that the captioning model finds it difficult to caption non-English images. This ablation study experiment is thus mostly exploratory, and more experiments are needed to assess the performance benefits coming from seeing more non-English images, versus seeing more diverse linguistic concepts in (translated) non-English captions.

# G  All DFN filtering baselines

Table 7: Here we report all the baselines we experiment with using the public DFN from [13] for filtering; all models are trained for 128M steps as set by the DataComp benchmark. For each caption distribution (i.e., raw/ translated/ English-only), only the filtering threshold that yields the best average performance across 38 tasks is shown in Table 1.

| Baseline name | Dataset size | ImageNet | ImageNet shifts | Retrieval | GeoDE | Average over 38 tasks |
|---|---|---|---|---|---|---|
| Top 20% raw captions | 25.6M | 0.316 | 0.260 | 0.282 | 0.688 | 0.350 |
| Top 20% raw captions, replaced with translated captions | 25.6M | 0.304 | 0.252 | 0.268 | 0.668 | 0.331 |
| Top 30% raw captions | 38.4M | 0.297 | 0.246 | 0.280 | 0.663 | 0.337 |
| Top 40% raw captions | 51.2M | 0.267 | 0.222 | 0.274 | 0.669 | 0.320 |
| Top 20% translated captions | 25.6M | 0.329 | 0.275 | 0.296 | 0.709 | 0.359 |
| Top 30% translated captions | 38.4M | 0.311 | 0.265 | **0.305** | 0.718 | 0.352 |
| Top 40% translated captions | 51.2M | 0.289 | 0.248 | 0.288 | 0.709 | 0.332 |
| Top 20% raw English-only captions | 8.0M | 0.260 | 0.218 | 0.234 | 0.603 | 0.303 |
| Top 30% raw English-only captions | 12.0M | 0.280 | 0.238 | 0.259 | 0.630 | 0.326 |
| Top 40% raw English-only captions | 16.0M | 0.283 | 0.236 | 0.278 | 0.666 | 0.327 |
| Top 50% raw English-only captions | 20.0M | 0.277 | 0.236 | 0.280 | 0.668 | 0.321 |
| Top 20% raw captions ∪ top 20% translated captions, using translated caption for all | 34.2M | 0.316 | 0.265 | 0.289 | 0.716 | 0.353 |
| Top 20% raw captions ∪ top 20% translated captions | 34.2M | 0.329 | 0.271 | 0.298 | 0.720 | **0.364** |
| Top 20% raw captions & top 20% translated captions | 51.2M | **0.336** | **0.280** | 0.301 | **0.725** | 0.361 |
| Top 30% raw captions & top 30% translated captions | 76.8M | 0.295 | 0.248 | 0.282 | 0.673 | 0.340 |

# H  Training for longer

Here we show the results of all the baselines that we train for 1.28B steps (i.e., 10× the number of steps set by DataComp). In Table 8, we find that when using either raw web-crawled captions or English-translated captions, filtering for top 30% of the pool does best, and translated multilingual captions continue to yield better performance on standard metrics compared to raw captions. We also provide a breakdown of performance differences between these two baselines across 38 tasks from DataComp in Figure 8.

Table 8: **When the training duration is increased by 10× compared to the DataComp setting, training on translated multilingual captions continues to outperform training on raw captions across a range of metrics; using both sources of captions continues to yield the best performance.** We show performance of all the baselines that are trained for 1.28B steps. Even though using filtered raw captions and using filtered translated captions yield similar average performance (0.414 percentage points), the latter still surpasses the former on ImageNet, ImageNet distribution shifts, retrieval and GeoDE (worst-region accuracy). We also note that these performance gaps widen with training duration (see Table 1 for a comparison).

| Baseline name | Dataset size | ImageNet | ImageNet shifts | Retrieval | GeoDE | Average over 38 tasks |
|---|---|---|---|---|---|---|
| Top 20% raw captions | 25.6M | 0.423 | 0.345 | 0.331 | 0.751 | 0.407 |
| Top 30% raw captions | 38.4M | 0.414 | 0.340 | 0.344 | 0.742 | 0.414 |
| Top 40% raw captions | 51.2M | 0.417 | 0.344 | 0.358 | 0.746 | 0.410 |
| Top 20% translated captions | 25.6M | 0.421 | 0.348 | 0.346 | 0.754 | 0.412 |
| Top 30% translated captions | 38.4M | 0.427 | 0.347 | 0.352 | 0.771 | 0.414 |
| Top 40% translated captions | 51.2M | 0.421 | 0.348 | 0.346 | 0.754 | 0.412 |
| Top 20% raw captions ∪ top 20% translated captions | 34.2M | 0.441 | 0.359 | 0.353 | 0.775 | 0.427 |
| Top 20% raw captions & top 20% translated captions | 51.2M | **0.456** | **0.369** | **0.371** | **0.776** | **0.435** |
| Top 30% raw captions & top 30% translated captions | 76.8M | 0.419 | 0.347 | 0.345 | 0.771 | 0.429 |

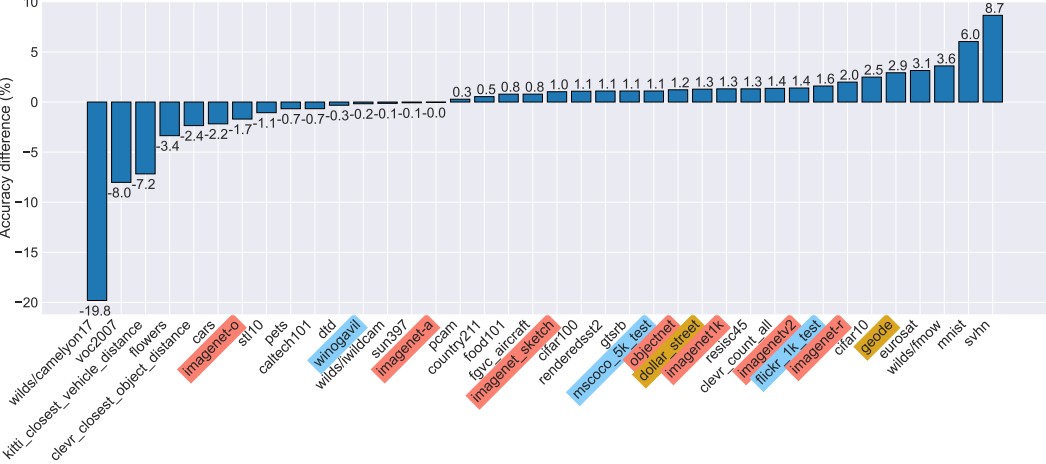

Figure 8: **With the same degree of filtering, training with (image, translated caption) pairs improves performance on 23 out of 38 tasks compared to training with (image, raw caption) pairs, including ImageNet distribution shifts, retrieval, and tasks with geographically diverse inputs.** We compare performance on each task of the DataComp benchmark between training with raw captions and training with translated captions, when both are trained for 1.28B steps. Both datasets have also been filtered with image-text cosine similarities output by the public DFN [13] to select the top 30% examples. We find that when we increase training duration to be 10× longer than DataComp's setting, using translated multilingual captions and using raw captions yield similar average performance across 38 tasks. However, the former still outperforms the latter on most of the ImageNet distribution shifts (red), retrieval (blue) and fairness-related tasks (dark yellow).

# I More performance analysis

In addition to the analysis in 4.3, we provide further breakdown of performance changes at the income group and class levels, on Dollar Street (Figure 9) and ImageNet (Figure 10) respectively, when we swap the training distribution from top 30% raw captions to top 30% translated captions.

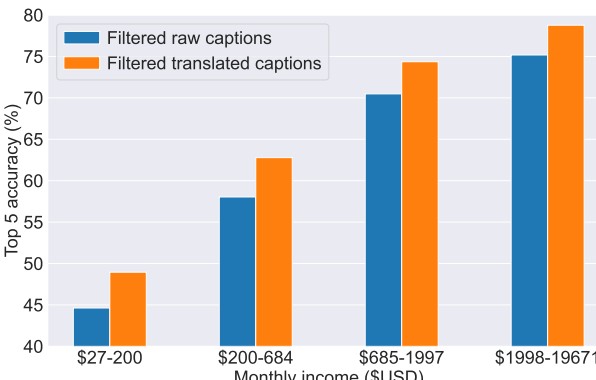

Figure 9: **On Dollar Street, using translated multilingual captions leads to performance improvement across all income groups.** Dollar Street [39] is another fairness-related task that involves classifying images of everyday items collected from households around the world with different socioeconomic backgrounds. We break down the performance on this dataset by income groups and find that training on top-quality translated captions improves the classification accuracy across all groups, compared to training on top-quality raw captions.

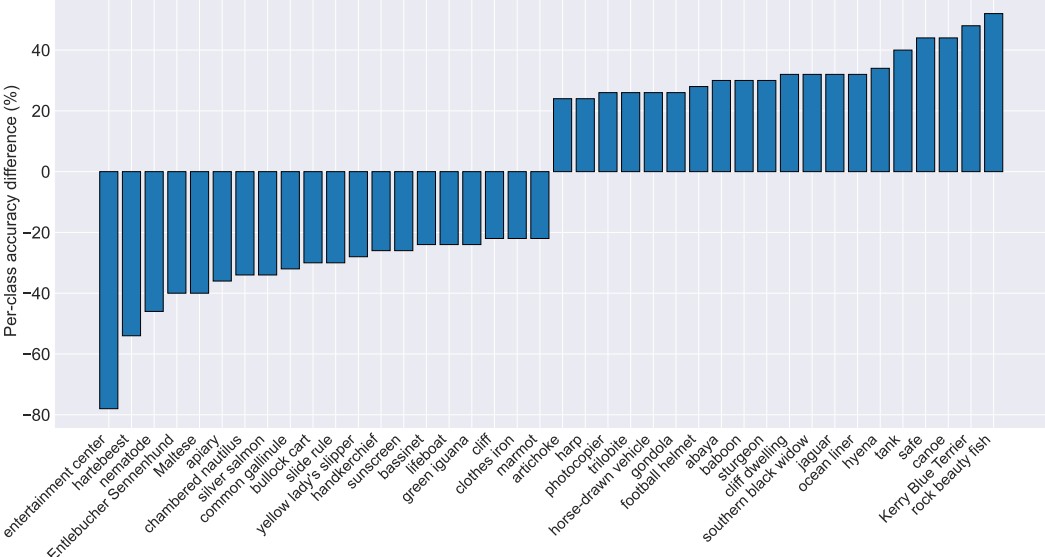

Figure 10: **40 ImageNet classes that observe the largest changes in classification performance when we train on top translated multilingual captions compared to top raw captions.** We show 40 categories from ImageNet that see the biggest change in accuracy when more (translated) multilingual data is included in the training set.

