# OpenReview forum: "Multilingual Diversity Improves Vision-Language Representations"
_NeurIPS.cc/2024/Conference — NeurIPS 2024 spotlight_

### Official Review · Reviewer_2aum · 2024-07-01

**Soundness:** 3
**Presentation:** 3
**Contribution:** 2
**Rating:** 5
**Confidence:** 4

**Summary:**

The paper conducts a systematic study to explore the performance benefits of using translated non-English data for English vision tasks. By translating multilingual image-text pairs from a raw web crawl to English and re-filtering them, the authors show that continual pre-training on this data increases the performance of the English model on a diverse set of tasks with a single model.

**Strengths:**

* The authors show that by incorporating translated multilingual and multicultural VL dataset can improve English model's results empirically on English only tasks.
* The paper provides interesting analysis showing that using diverse training data improves benchmark results that require geographical or cultural diversity.

**Weaknesses:**

- The improvement from using translated data diminished as the training converged, which is not surprising. Similarly, using more training data to improve the evaluation results on downstream tasks is also expected. Hence, the novelty of insights is limited in this paper.
- The paper only evaluated on one LLM, which more evaluations are required to validate the generality of the claim of this paper.

**Questions:**

Possible experiments to make the paper better:
- Add more multicultural evaluation datasets
- Add more evaluations to show the benefits of using diverse training data could benefits multiple models sizes and types
- Perform similar training and evaluation in other languages, to show all languages would benefits from diverse training images and captions

**Limitations:**

While this research may be interesting for those working primarily on monolingual research (i.e., English), its framing should be more sensitive to the many efforts related to multilingual and multicultural NLP. Personally, I think the benefits of using diverse images and captions can be achieved by collecting diverse images and paraphrasing captions or through better sampling of the training data, rather than just using multilingual or multicultural datasets and translating them. As a community, we already lack people working on multicultural and multilingual models, yet this work uses data and research from other cultures to improve **English-only** models (although with good intentions). Perhaps there could be some discussions on this.

---

> ### Author Rebuttal · Authors · 2024-08-07
>
> We thank the reviewer for the feedback.
>
> It is very important that we clarify a misconception: our work deals with vision-language models and vision benchmarks. We do not train any LLMs or make claims about the state of multilingual NLP research.
>
> **Framing should be more sensitive to multilingual efforts in NLP/ This work uses data to improve English-only models while we already lack people working on multilingual models.** To clarify,
>
> (i) We support multilingual and multicultural efforts. While benchmarking and improving performance on multilingual tasks is not the focus of our work, as one step in this direction, we enable English-language models to gain exposure to more samples of diverse origins and consequently, do better on geographically diverse benchmarks (e.g. GeoDE and DollarStreet). We discuss how to adapt models from our paper to further enhance their multilingual capabilities under the “Future work” section. Comparing the multilingual performance of these post-training adaptation methods to directly training on non-English captions merits a separate investigation.
>
> (ii) Instead, the goal of our work is to challenge the status quo: many ML practitioners seek to improve state-of-the-art performance on popular benchmarks, which happen to be English vision tasks like Imagenet. In the process of overfitting to these benchmarks, previous work discards a lot (if not all) of non-English samples due to the belief that non-English data does not benefit English evaluations. We question this practice. But in order to *incentivize change*, we need to show that it is possible to improve the English-centric “state-of-the-art” by using more data of non-English origins. We hope that our positive findings could help multilingual and multicultural efforts become the default design choice for data curation, instead of existing as a second priority or as a separate (societal) consideration.
>
> **The improvement from using translated data diminished as the training converged, which is not surprising.** We want to clarify that the results we reported show the opposite. From Table 1 of the main paper, our best baseline, “Filtered raw captions & Filtered translated captions”, outperform “Filtered raw captions” by 2.0% on ImageNet and 1.1 percentage points on average across 38 tasks. When training for 10x more steps, the performance gaps increase to 4.2% and 2.1 percentage points respectively.
>
> **Using more training data to improve the evaluation results on downstream tasks is also expected.** For our baselines, we fix the training budget and the amount of raw data available (i.e. initial pool). With each caption distribution, we tune the filtering threshold, and consequently the size of the resulting filtered subset used for training (this was reported in Table 1 of the main paper). We note that with this setup, increasing *filtered data quantity* is *not* guaranteed to lead to better performance on downstream tasks, as larger subset (with less strict filtering) means fewer passes through each datapoint. In Table 1 we only report the *best performance obtainable* after filtering each data distribution. Table 6 (Appendix) contains the full results, accounting for the number of samples used. While “Top 20% raw captions & Top 20% translated captions” indeed uses more training data than “Top 20% raw captions” (51.2M versus 25.6M), when controlling for data quantity taken from the latter distribution, “Top 20% raw captions & Top 20% translated captions” still significantly outperforms “Top 40% raw captions” (both having 51.2M samples).
>
> **Add more multicultural evaluation datasets.** Given the reviewer’s concern, we have added a new evaluation on Google Landmarks dataset (Weyand et al., 2020). When training for 10x longer, using filtered raw captions yields 16.9% classification accuracy, while our best baseline (using a mix of filtered raw captions and translated captions) gets 18.5% accuracy.
>
> **Show the benefits of using diverse training data could benefit multiple models sizes and types, and other languages.** Due to compute constraints, we did not experiment with more model types and translating to other languages, but we will add these points to our Discussion section. For context, obtaining the baseline numbers for our paper (see Appendix Tables 6 & 7) took about 850 GPU hours with 8 A40 GPUs. Besides, we find that extending our approach to other languages (e.g. translating all captions to Chinese and measuring performance on Chinese tasks) is sufficient scope for a separate paper.

---

> > ### Comment · Reviewer_2aum · 2024-08-12
> > **Re**
> >
> > Thank you very much for the additional clarifications. I have revised the score to reflect my current assessment of the paper.

---

> > > ### Author Response · Authors · 2024-08-13
> > > **Author Response**
> > >
> > > Thank you for acknowledging our rebuttal. As mentioned in our comment, we are happy to engage in further discussion at any point in the future as well.

---

### Official Review · Reviewer_7AUA · 2024-07-12

**Soundness:** 4
**Presentation:** 4
**Contribution:** 3
**Rating:** 8
**Confidence:** 4

**Summary:**

This paper points out an important issue in current training of CLIP models---the push for including more english-centric / english-only data  in the pretraining dataset. The paper points out that this is mainly driven by the downstream evaluation test-beds primarily being english-focused and hence the need to include multi-lingual data has not been found so far. The paper conducts a thorough study of using multilingual data into the pretraining datasets of CLIP models by using caption translation models. The paper evaluates and shows improvements on several downstream datasets from the 38-Datacomp benchmark evaluation suite, and provides analysis on the key differences between multilingual and english-only pretraining datasets.

**Strengths:**

- The paper is very well written and easy to follow
- All the claims in the paper are verified with substantial empirical evidence
- The paper's main message is adequately represented throughout the paper without any distracting claims, hence all the experiments done in the paper flow very smoothly
- The paper points out a very important issue in current pretraining datasets of CLIP models, and showcases a simple intuitive fix for future models and pretraining datasets

**Weaknesses:**

- The paper could benefit from some additional analysis. I note down some analayses that I think might strengthen the paper below.

- For a particular concept is it possible to showcase the diversity afforded by looking at only English data and multilingual data? For instance, take the concept of “stove”. You could perhaps manually take 10-30K images of a stove (assuming that many exist), by filtering the texts for this particular concept, and do the same for both English only and multilingual data. Then, quantify the diversity of images, by perhaps taking a strong Dino-v2 pretrained encoder and then clustering them, and perhaps looking at some intra- and inter-cluster distance metrics. I think this would nicely validate the qualitative visualisation in figure 1 with some empirical evidence that multilingual data indeed boosts the diversity of visual concepts in pretraining datasets. It would be great to see this analysis on perhaps one or two concepts.
Why is this an important analysis? My reasoning would be that while it is totally plausible that multilingual concepts add diversity, it is unclear to me how much of this is actually true? Since it is plausible that English webpages also contain such diverse visual concepts, I think it would be nice to see if there are “new visual concepts” being included in the pretraining corpus, or is it mostly that these “culturally-diverse” visual concepts were present in the english-only pretraining datasets, but we just boost up their frequency in the pretraining dataset? I think either of the two cases is a valid justification for the claim that multilingual data increases diversity, but it would be nice to have a precise answer for what exactly out of the two is the case.

- The main claims of the paper that multilingual data improves VL representations is sufficiently backed up, but benchmarking on culturally diverse / multilingual-sourced data is still limited. The paper only considers GeoDE and DollarStreet for evaluation. I would recommend to also benchmark and report the performance on some more culturally and geospatially diverse datasets like [1,2,3,4]. Some of these datasets might need to be ported into a classification / retrieval format, but in general I think this is an important experiment to do to further validate the important of multilingual-sourced pretraining data.

- By filtering on top of the translated image-text pairs, it seems likely that the training data diversity increases, and potentially train-test similarity [5] also increases. This could be an added confounder in the takeaways of the paper. Could the authors comment/discuss this a bit more in the paper?

- The analysis in the paper would further be improved by checking how the distribution of CLIPScores look before and after the translation. Potentially there would be a shift to the right in the similarity distribution, but it would be a good analysis to include for better intuitions on what might be good filtering thresholds if such a model were to be used in the future for vector arithmetic / data filtering itself.

[1] Kalluri et al, GeoNet: Benchmarking Unsupervised Adaptation Across Geographies
[2] Weyand et al, Google Landmarks Dataset v2 -- A Large-Scale Benchmark for Instance-Level Recognition and Retrieval
[3] Yin et al, Broaden the Vision: Geo-Diverse Visual Commonsense Reasoning
[4] Romero et al, CVQA: Culturally-diverse Multilingual Visual Question Answering Benchmark
[5] Mayilvahanan et al, Does CLIP’s generalization performance mainly stem from high train-test similarity?

**Questions:**

I have a few additional questions/comments which I include below:

- How do you estimate the english part of the web-crawl? You mention it is in 1/3rd in the introduction.

- Is there an explanation for why filtered translated captions might improve english-centric performance over the filtered raw captions? This seems counter-intuitive given as you say in the paper, that ImageNet is mostly western-centric. This also seems to contradicts the results of the No-Filter paper [1].

- Could the model trained on this filtered raw subset itself be used as a DFN, and potentially further improve the performance when training on the raw multilingual pool? The idea being that you might have to do less tuning on the filtering threshold since the model is more robust to noise from the translated captions.

- In fig 3, are the results from the "filtered raw captions" and "filtered translated captions" methods from tab 1? Doesn’t seem so since the performance improvements on the 38 tasks mentioned in the caption of fig 3 is 1.5% whereas the performance improvements from tab 1 is only 0.9%.

- Why does performance on Food101 go down? Is it because all the food classes included in Food101 are primarily English-centric? I am certain that is not the case. I would be very interested in a more detailed analysis of why the performance on Food101 goes down when we include more multilingual data since that is one aspect that I would have thought improves quite a lot.

[1] Pouget et al, No Filter: Cultural and Socioeconomic Diversity in Contrastive Vision-Language Models

**Limitations:**

The authors are explicit about their limitations. Any further limitations I think are mentioned in the weaknesses section above.

---

> ### Author Rebuttal · Authors · 2024-08-07
>
> We thank the reviewer for all the valuable suggestions!
>
> **Showcase the diversity for certain concepts.** Per your suggestion, we ran additional analysis to compare the diversity of images for the same concept. We sampled 1K images from each data distribution for which the corresponding (translated) captions mention “stove”. With DINOv2, we obtained embeddings of these images and used them to cluster each data distribution into 20, 50 and 100 clusters. Figure 2 in our global response shows the average inter-cluster distance of “stove” images in English data versus non-English data. Overall the latter group of images yields higher inter-cluster distance, suggesting that the clusters are more well-separated and the non-English “stove” images are more diverse. We will add this experiment to the paper.
>
> **Benchmarking on culturally diverse data is still limited.** Our work does not explicitly seek to improve performance on culturally diverse tasks. We explore whether increasing the diversity of the data origins, and consequently allowing the training set to be dominated by non-English samples, can improve performance on vision tasks *in general*, especially on those that dictate the field’s state-of-the-art like ImageNet. Improvement on geographically diverse benchmarks is an added benefit in this process. However, given the reviewer’s feedback, we have added a new evaluation on Google Landmarks dataset (Weyand et al., 2020). When training for 10x longer, using filtered raw captions yields 16.9% classification accuracy, while our best baseline (mixing filtered raw captions and translated captions) gets 18.5% accuracy.
>
> **By filtering on top of the translated image-text pairs, potentially train-test similarity also increases.** Changes in train-test similarity may happen as a result of using translated captions, but that should not be a confounder in the paper takeaways because (i) the proposed translation method does not directly optimize for train-test similarity, (ii) we apply the same filtering process (i.e. based on DFN score) to both raw and translated captions, (iii) we measure performance on a large number of vision tasks (38).
>
> **Checking how the distribution of CLIPscore look before and after the translation.** We have added this analysis in our global response (see Figure 1 of the attached PDF). We sampled 10K images with English captions and 10K with non-English captions from the initial pool, and compared how the DFN scores change with translation. Unsurprisingly, DFN scores for non-English samples generally increase after the captions are translated into English. This in turn leads to a right shift in the distribution of image-text similarity scores when looking at the score histograms before and after translation. As image-text alignment (measured by DFN score) tends to help with empirical performance, this shift suggests that translation increases the availability of good training data.
>
> **Estimate that English takes up ⅓ of any web crawl.** This estimate is computed from using the NLLB model to detect languages in the DataComp dataset. Since DataComp performed minimal preprocessing (i.e. NSFW removal and test set deduplication), we hypothesize that the initial DataComp pool is representative of the natural distribution of raw data on the web that is appropriate for training.
>
> **Improvement on ImageNet contradicts the results of the No-Filter paper.** We note some differences between our experiment setup and that of concurrent work:
>
> (1) We compare training on filtered translated captions (which is *dominated* by non-English data) to training on filtered raw captions (which is *dominated* by English data). In contrast, the No-Filter paper compares multilingual data (globe) to its strict subset containing *only* English data (en).
>
> (2) We perform filtering and retain only a small fraction of the original pool (20-30%), whereas the No-Filter paper trains on web data with “minimal filtering”. We posit that filtering is important to obtain competitive performance with translated captions, especially given the noise introduced by the translation process.
>
> We believe our setup better mimics how pre-training data curation is commonly done in practice (e.g. LAION, which was also heavily filtered and contains a mix of English and non-English data).
>
> **Why filtered translated captions might help improve English-centric performance.** While the fraction of images of English origins is lower when we use filtered translated captions instead of filtered raw captions (60% → 40%, Figure 2 of our paper), the number of such images in the final training set is still significant. As a result, the model can still learn English-centric concepts to a certain extent. We hypothesize that the performance on English vision tasks is further reinforced by training on high-quality, diverse multilingual data, which helps induce better visual features and robustness in general.
>
> **Average performance improvement in Figure 3.** The analysis in Figure 3 of our paper is performed on models trained with the top 30% of the initial pool (see description in Lines 251-254), whereas the first section of Table 1 involves training on the top 20% of the pool. We pick the 30% threshold for further analyses because (i) it is the baseline that yields the best performance when training until convergence, (ii) the average performance gap between filtered translated captions and filtered raw captions is larger at this threshold, allowing us to observe the differences on individual tasks better.
>
> **Why does performance on Food101 go down?** We note that (1) the accuracy change for the Food101 task between using filtered translated caption and using filtered raw captions is relatively small (-0.2%), and (2) the Food101 class names are still dominated by English-centric concepts. Besides, the noise in the translation process may affect whether the class mentions in the original languages are still preserved after translation.

---

> > ### Comment · Reviewer_7AUA · 2024-08-10
> > **Response to author rebuttal**
> >
> > I thank the authors for their thorough analysis and further experiments.
> >
> > - The clustering experiment now adds a very strong demonstration that the non-English data improves the diversity of visual samples even for concepts that are already present in the English pool. This is a great result!
> >
> > - GLD experiments look great and provide additional validation for the significance of including non-english data.
> >
> > - The points discussing the differences between your and the No-Filter paper are quite valid and important. I would encourage the authors to add this as an explicit discussion section either in the related works or the appendix.
> >
> > After reading all the other reviewer comments and the author's responses to them, I am inclined to strong accept this paper as it makes a significant contribution to the data-centric VL community and provides relevant, practical insights. I am increasing my score to 8.

---

> > > ### Author Response · Authors · 2024-08-12
> > > **Follow up**
> > >
> > > Thank you for acknowledging our additional results and for adjusting the rating! We will make sure to include the new analyses as well as a discussion of the No Filter paper results in the next version of our work.

---

### Official Review · Reviewer_9sUx · 2024-07-14

**Soundness:** 4
**Presentation:** 4
**Contribution:** 4
**Rating:** 8
**Confidence:** 3

**Summary:**

The authors investigate whether *multilingual* vision-language data improves the *English-only* performance on a model in vision-language tasks.

They translate captions from DataComp from English to other languages and train a CLIP model on these multilingual captions.

They find that this action boosts performance on both English-only tasks like imagenet matching and geographically diverse tasks like GeoDE.

**Strengths:**

This is a super efficient way to gain a performance boost on DataComp (for revs who might not be aware, this is a cool benchmark for assessing efficient training of vision-language representations evaluated over a bunch of downstream datasets within a fixed compute budget). Any method that boosts DataComp performance across so many tasks for free (just translation instead of collecting new captions) is awesome in my opinion.

Performance boost on GeoDE (geographically sorted img classification) is improved considerably in all regions, justifying the purpose of the intervention.

**Weaknesses:**

Unclear how the proposed automated translation pipeline ties in to the original motivation (culturally specific items not being captured in English data)

Could benefit from a bit deeper description of what the GeoDE task is for unfamiliar readers to improve reach.

**Questions:**

Figure 1: where are these examples from? Are they actually in the datacomp datasset? Doesn't translating datacomp captions into other languages keep the learned representations in the "English concept space" of sorts?

**Limitations:**

Yes, limitations are sufficiently covered.

---

> ### Author Rebuttal · Authors · 2024-08-07
>
> Thank you for the review and for recognizing the strengths of our work!
>
> **How the proposed automated translation pipeline ties in to the original motivation (culturally specific items not being captured in English data).** We discuss the presence of culturally salient concepts in the Introduction mainly to motivate how multilingual data is inherently enriching and complements English data. Improving the availability of culturally specific items in the final training set is not what our method specifically optimizes for, but is likely to happen with the inclusion of substantially more samples of non-English origins after filtering (see Figure 2 of main paper).
>
> **Deeper description of what the GeoDE task is for unfamiliar readers.** Thank you for raising this point. GeoDE is a geographically diverse dataset containing 61K images spanning 40 common object categories, collected from 6 different world regions via crowd-sourcing. We will add more details of this task to the paper.
>
> **Where are the examples in Figure 1 from? Are they actually in the datacomp dataset?** Yes, these examples are taken from our training data, i.e. DataComp.
>
> **Doesn't translating datacomp captions into other languages keep the learned representations in the English concept space?** We interpret this question as whether translation would reduce the richness of the concept space (especially when it comes to translating culturally salient concepts). We empirically observe that after translating multilingual captions to English, some culturally salient concepts are still preserved - refer to Figure 1 (left) of the main paper for examples. Our work has also acknowledged this potential limitation, that “translation can sometimes be too literal, subject to losing the intent and richness of the original phrasing”. Nevertheless, we hope that our findings inspire future work to (i) be more inclusive of data of non-English origins during training, (ii) investigate other ways to effectively leverage the diversity of multilingual data.

---

> > ### Comment · Reviewer_9sUx · 2024-08-13
> >
> > **Motivation of translating from English**
> >
> > This explanation is reasonable. I'd like to see this clarified in the intro of the CR.
> >
> > **Other answers**
> >
> > Reasonable, thank you for clarifying. I forgot that datacomp pre-filtering does indeed contain multilingual data.
> >
> > **Translating datacomp captions**
> >
> > After the clarification, I understand. You are indeed augmenting the concept space by translating into English here.
> >
> > Thank you for your responses, I will update my soundness score.

---

> > > ### Author Response · Authors · 2024-08-13
> > > **Author response**
> > >
> > > Thank you for acknowledging our rebuttal and for adjusting the rating! We are glad to hear that our response has provided clarification to your earlier questions.

---

### Official Review · Reviewer_xoEn · 2024-07-15

**Soundness:** 3
**Presentation:** 3
**Contribution:** 3
**Rating:** 7
**Confidence:** 4

**Summary:**

In this paper, the authors conduct a thorough exploration of how multilingual image-text pairs benefit English vision tasks. They first present how to effectively utilize translated data to improve performance on standard vision tasks and derive valuable conclusions through detailed ablation studies. Second, they illustrate the differences in image and text distributions between English and non-English image-text pairs. These findings highlight the potential of leveraging multilingual datasets to enhance the robustness and accuracy of vision models.

**Strengths:**

1. The paper presents effective strategies for utilizing translated data to enhance performance on standard vision tasks
2. Detailed ablation studies are carried out to derive valuable conclusions, which help the community understand the impact of various factors.
3. The paper is well-written and easy to follow.

**Weaknesses:**

1. No quantitative assessment for the quality of translation.
2. Only evaluation on representation task. Does such data strategy also work for the training of Multimodal LLM?

**Questions:**

No

**Limitations:**

Yes

---

> ### Author Rebuttal · Authors · 2024-08-07
>
> We thank the reviewer for their time and feedback!
>
> **No quantitative assessment for the quality of translation.** Given your concern, we have performed additional analysis of the translation quality. We sampled 100K multilingual captions from our raw data pool and backtranslated the English-translated caption into the original language (e.g. Chinese text -> English translation -> Chinese translation of the English-translated text). Then we computed the cosine similarity between the initial web caption and the backtranslated caption using embeddings from the multilingual Sentence-BERT model (Reimers et al., 2019), to assess how much semantic meaning is preserved after translation. We find that on average the cosine similarity (thus, translation quality) remains relatively high (0.63). Below we report the top 5 and bottom 5 languages that observe the highest and lowest translation quality as captured by our metric (each has at least 30 samples). We will add the full analysis in the next version of the paper.
>
> | Language | Text cosine similarity after backtranslation |
> | -------- | ------- |
> | English | 0.886 |
> | Norwegian Nynorsk | 0.883 |
> | Bengali | 0.883 |
> | Russian | 0.860 |
> | Norwegian Bokmål | 0.839 |
> | | |
> | Marathi | 0.271 |
> | Irish | 0.240 |
> | Standard Latvian | 0.233 |
> | Chechen | 0.0595 |
> | Karachay-Balkar | 0.00280 |
>
> **Only evaluation on representation task. Does such data strategy also work for the training of multimodal LLM?** We note that our evaluations already contain 38 datasets involving recognition and classification of a wide range of domains (e.g. texture/ traffic sign/ scene/ metastatic tissue/ geolocation/ animal/ etc.), in addition to image-text retrieval and commonsense association.
>
> Extending the findings from this work study to multimodal LLM training would be an interesting direction for future work, but is orthogonal to our contribution; we will add this to the Discussion section. Our results demonstrate that there is so much diversity and visual information to be leveraged from multilingual data, to the extent that training on predominantly non-English samples can improve CLIP’s performance on standard English benchmarks. We hypothesize that leveraging this diversity will also be beneficial for training multimodal LLMs, especially since many of which still rely on English-dominated image-text datasets (e.g. LAION) and still use CLIP as the image encoder (e.g. LLaVA-1.5).

---

### Author Rebuttal · Authors · 2024-08-07

We would like to thank all reviewers again for providing thoughtful reviews of our work. Here we highlight several new results/ analyses taking your feedback into consideration:
\
&nbsp;
1. Reviewer xoEn wondered about the translation quality. In response, we sampled 100K captions from our raw data pool and backtranslated the English-translated caption into the original language (e.g. Chinese text -> English translation -> Chinese translation of the English-translated text). To assess the translation quality, we computed the cosine similarity between the initial web text and the backtranslated text using embeddings from the multilingual Sentence-BERT model (Reimers et al., 2019). We find that on average the cosine similarity (and thus, translation quality) remains relatively high (0.63). Below we report the top 5 and bottom 5 languages that observe the highest and lowest translation quality measured by our metric:
| Language | Text cosine similarity after backtranslation |
| -------- | ------- |
| English | 0.886 |
| Norwegian Nynorsk | 0.883 |
| Bengali | 0.883 |
| Russian | 0.860 |
| Norwegian Bokmål | 0.839 |
| | |
| Marathi | 0.271 |
| Irish | 0.240 |
| Standard Latvian | 0.233 |
| Chechen | 0.0595 |
| Karachay-Balkar | 0.00280 |

2. Reviewer 7AUA asked about DFN score changes before and after translation. We analyzed the score differences for 10K English and 10K non-English samples, and found that translation improves the image-text similarity score of non-English samples, as well as the overall score distribution of the data pool. Refer to Figure 1 of the attached PDF for more details.
\
&nbsp;
3. Reviewer 7AUA suggested adding another metric for quantifying the diversity of multilingual data by measuring clustering distances. In Figure 2 of the response PDF, we showed that for a specific concept such as “stove”, non-English images form more distinct clusters than English ones. We randomly sampled 1K images with English captions and 1K with non-English captions, such that the (translated) captions mentioned “stove”, and embedded them with the DINOv2 model. Across different numbers of clusters uncovered, non-English data generally yields higher inter-cluster distance, suggesting that the “stove” images with multilingual captions are more heterogeneous compared to those with English captions.
\
&nbsp;
4. Since Reviewers 2aum and 7AUA requested more multicultural evaluations, we ran a new evaluation on Google Landmarks dataset (Weyand et al., 2020). When training for 10x longer, our best baseline (using a mix of filtered raw captions and translated captions) yields 18.5% classification accuracy, outperforming the baseline trained on just filtered raw captions by 1.6%. We note that the performance on this task is limited given that our best data mix only uses ~51M samples.
\
&nbsp;
5. Last but not least, Reviewer 2aum had concerns about our investigation not being sensitive to the many efforts related to multilingual and multicultural NLP. **Here we would like to restate the goal of our work:**

We advocate for increasing the cultural and linguistic diversity of the training data for vision-language models (VLMs), and view our findings as complementary to other multilingual and multicultural efforts. While these efforts have led to new geographically diverse benchmarks as well as new ways to increase the availability of high-quality non-English data, their adoption is still relatively limited. This is evident from the fact that many popular models that are considered state-of-the-art (e.g. MetaCLIP, SigLIP, ALIGN) were still trained on entirely English image-text pairs. In order to change this status quo, it is important to demonstrate that training on predominantly data of non-English origins can do better than training on predominantly English data, especially on standard vision tasks that define the field’s state-of-the-art (e.g. ImageNet). Our work provides such evidence. We hope this can lead to more active exploration and adoption of multilingual and multicultural data in mainstream VLM training, instead of using this data only when under-served populations or tasks are involved.

---

### Author Response · Authors · 2024-08-12
**Rebuttal follow-up**

We thank reviewers again for their time. As the discussion period is drawing to a close, we would like to ensure we have addressed all of the concerns. Let us know if you you have any further questions/ thoughts after our rebuttal, we would be happy to engage in further discussion.

-- Paper authors

---

### Decision · Program_Chairs · 2024-09-25

**Decision:**

Accept (spotlight)

**Comment:**

This paper presents a conceptually simple but extensive studies of how translated multilingual (non-English) image-text pairs benefit English vision tasks. They also provide an intriguing analysis of the differences in image and text distributions between English and non-English image-text pairs, supporting the claim that multilingual data enhance diversity.

Overall, all reviewers were favorable to the unique focus and findings of the paper, and were satisfied with the authors' rebuttal. As a result, their final scores are unanimously toward acceptance, with strong supports from three reviewers. The AC would like to support their judgement.